


# Negative Differential Resistance, Instability, and Critical Transition in Lightning Leader

**Xueqiang Gou[1]†, Chao Xin[1] Liwen Xu[1] Ping Yuan[1] Yijun Zhang[2] Mingli Cheng[3]**

[1] College of Physics and Electronic Engineering, Northwest Normal University, Lanzhou 730070, China

[2] Department of Atmospheric and Oceanic Sciences & Institute of Atmospheric Sciences, Fudan University, Shanghai, 200438, China

[3] Department of Building Environment and Energy Engineering, The Hong Kong Polytechnic University Hong Kong, SAR, China

†*Corresponding to*: Xueqiang Gou (1491168405@qq.com)

**Abstract.** The phenomena of leader extinction and restrike during lightning events, such as multiple strokes in ground flashes or recoil leaders in cloud flashes, present significant challenges. A key aspect of this issue involves the discussion of the channel's negative differential resistance and its instability. From the perspective of bifurcation theory in nonlinear dynamics, this paper posits an inherent consistency among the channel's negative differential resistance, channel instability, and the critical transition from insulation to conduction. This study examines the differential resistance characteristics of the leader-streamer system in lightning development. We correlate the differential resistance characteristics of the leader-streamer channel with the channel's state and instability transitions, investigating the critical current and potential difference conditions required for the stable transition of the leader-streamer channel.

**Key words:**

negative differential resistance; instability; critical transition;lightning

## 1 Introduction

Natural lightning exhibits an intermittent nature distinct from long-gap discharges observed in laboratory settings (Gou et al., 2010; Gou et al., 2018; Iudin et al., 2022). This intermittency is closely related to the fractal and critical characteristics of the lightning process (Bulatov et al., 2020; Sterpka et al., 2021; Iudin & Syssoev, 2022; Syssoev et al., 2022). Additionally, the asymmetry between positive and negative polarities introduces inherent instability in the discharge process, leading to destabilization and re-excitation of lightning events (Van der Velde & Montanya, 2013; Williams, 2016; Williams & Heckman, 2012; Iudin, 2021; da Silva et al., 2023; Scholten et al., 2023).



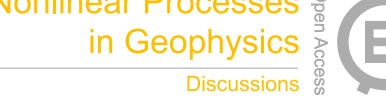

In ground flashes, negative ground flash discharges are typically separated by long
intervals of dim light. When the downward negative channel decays and eventually stops, the
active intracloud positive part intermittently generates conditions for the formation of dart or
dart-stepped leaders (Van der Velde & Montanya, 2013; Stock et al., 2014; Stock et al., 2023;
Lapierre et al., 2017; Jensen et al., 2023). In cloud flashes, the active positive leader contrasts
sharply with the longer intervals of the negative leader, where K-processes or recoil leaders in
the cloud are similar to dart or dart-stepped leaders in negative ground flashes (Van der Velde &
Montanya, 2013; Stock et al., 2014; Lapierre et al., 2017).
Recoil leaders are generally believed to arise from instability in the bidirectional leader
channel due to current interruption (Williams & Heckman, 2012; Mazur, 2016). Stepped leaders
are thought to emerge from various instabilities within the streamer channel at the leader's end
and their critical transitions (Malagón-Romero & Luque, 2019; Hare et al., 2021). These
instabilities often manifest as the negative differential resistance characteristics of the channel.
In gas discharge physics, negative differential resistance is typically associated with
bistability, hysteresis, and unstable transitions (Bosch & Merlino, 1986; Lozneanu et al., 2002;
Agop et al., 2012; Raizer & Mokrov, 2013). Since the interaction between the leader and
streamer in lightning discharges is inseparable, this paper extends the study of negative
differential resistance properties from the leader to the leader-streamer system in lightning. It
investigates the stability and instability of the channel during lightning development and the
current and potential differences required to maintain channel stability.
**2. Method**
**2.1 Negative differential resistance in lightning**
Lightning, as a natural phenomenon of large-scale arc discharge, exhibits the
characteristic of negative differential resistance in its channel (Heckman, 1992; da Silva et al.,
2019). This means that as the current increases, the temperature and conductivity of the channel
also increases, leading to a further increase in current, while the internal electric field required to
maintain the current decreases, and the voltage across the channel decreases. In other words, an
increase in current leads to a decrease in voltage, and vice versa ($dV/dI < 0$). Krehbiel et al.
(1979) pointed out that the instability of negative differential resistance in the channel might be
the main reason for channel attenuation.





Heckman (1992) and Williams & Heckman (2012) conducted detailed studies on the
relationship between negative differential resistance and the multiplicity of negative ground
flashes. They suggested that although the negative differential resistance channel connected to
the extended streamer source (which can be considered a current source) is unstable, the
existence of resistance and capacitance in the channel itself (both in parallel) forms a stabilizing
factor. If the electrical response time constant RC for the channel resistance is greater than the
thermal attenuation constant τ of the channel, the channel is stable; otherwise, it is unstable. The
critical stability current is approximately 100A (Heckman, 1992; Williams, 2006; Williams &
Heckman, 2012).
Mazur & Ruhnke (2014) and Mazur (2016a, b) pointed out that equating the leader
channel to a parallel arc resistance and capacitance connection might not be appropriate. As the
characteristic of the negative effect of channel resistance exists over the entire range of lightning
currents, channel stability is not necessarily related to its negative differential resistance
characteristics. What determines the stability of the channel is the minimum potential difference
condition of the streamer zone at the channel tip (Bazelyan & Raizer, 2000; Mazur, 2016a).
Since the initiation and development are guided by a large number of streamers originating from
the front, the high resistance of the streamer zone is important for the channel's stability. We
suggest that the differential resistance properties of the lightning channel should not only be
determined by the leader channel but should also include the streamers at the ends of the leader
channel.
**2.2 Negative differential resistance and bistability**
Theoretical exploration of negative differential resistance, hysteresis, and bistability led to
the derivation of a normalized relationship between current $J$ and voltage $\phi$ in the discharge
channel (Agop et al., 2012):
$$\varphi = J\left(1 + \frac{a}{1+J^2}\right) \qquad\qquad (1)$$
Figure 1 shows the theoretical dependence of the normalized current on the normalized
voltage. It can be seen that as the parameter $a$ increases, the system changes from stable to
unstable. For example, when the parameter $a = 6$, the current monotonically increases with the
voltage, and the system is monostable. When the parameter a = 18, the system exhibits obvious



instability and bistability. Initially, the current increases slowly with the voltage and maintains a
certain state (section AB), but when the voltage reaches a certain limit (point B), the current
suddenly jumps, and the system transitions to a completely different state (point C). When the
voltage begins to decrease in this state, the current decreases with it but maintains a steady state
(section CD). When the voltage decreases to a certain value (point D), the current drops
suddenly, and the system returns to its original state (point A). As the voltage varies, the system
jumps back and forth between two different stable states, thus showing hysteresis, demonstrating
the system's stability,instability and their critical transition under different parameter conditions.

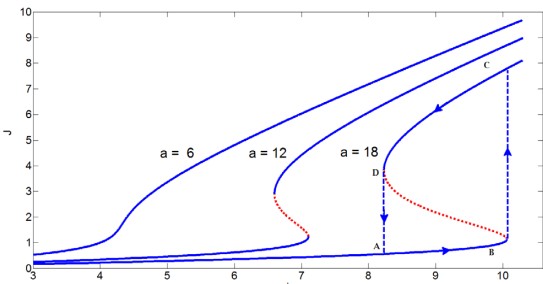


Fig. 1. Theoretical dependence of the normalized current on the normalized potential (adapted from Agop et
al., 2012, reprinted with permission from the Physical Society of Japan).
When examining nonlinear dynamics, it is not uncommon to observe negative differential
resistance, bistability, and hysteresis. By considering the dynamic system $\dfrac{dJ}{dt} = f(\phi, a, J)$,
where $J$ is the state variable and $\phi$, $a$ is a parameter, we can discern that the system is unstable
when $f(\phi, a, J)=0$ and $f'_J(\phi, a, J) > 0$, conversely, the system achieves stability when
$f(\phi, a, J)=0$ and $f'_J(\phi, a, J) < 0$ .if we let

$$f(\varphi, a, J) = \varphi - J\left(1 + \frac{a}{1+J^2}\right) \quad\quad (2)$$

then

$$f'_J(\varphi, a, J) = -1 - a\frac{1-J^2}{(1+J^2)^2} \quad\quad (3)$$



Considering equation (1), we have $f'_J(\phi, a, J) = -\phi'(J)$, then the system is unstable
under the condition $\phi = J(1 + \dfrac{a}{1 + J^2})$ and $\phi'(J) < 0$, in agreement with the previous result on
the instability of the negative differential resistance. The sign of channel differential resistance
provides insight into the stability of channel states and transitions of lightning.
Similar bistability, hysteresis, and critical transitions are widely observed in biological,
atmospheric, ecological, and other systems and can be described by similar dynamical systems
(Scheffer & Carpenter, 2003; Scheffer, 2009). The generation of instability and bistability can be
illustrated by the rolling ball model shown in Figure 2, where the peaks and valleys represent
unstable and stable points, respectively. Instability triggered by strong nonlinearities (positive
feedback) is an important factor causing the bistability (polymorphism) of the system and the
critical transition.

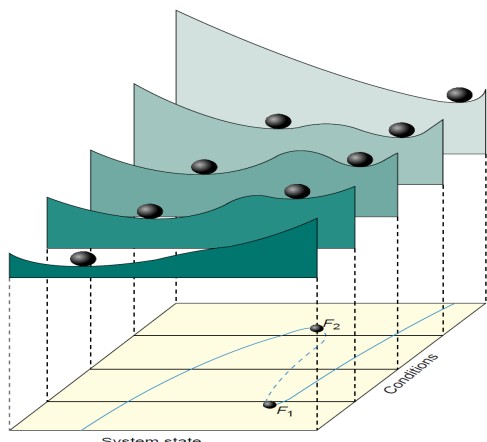


Fig. 2. Schematic representation of the locus of stability as a function of external conditions (adapted from
Scheffer & Carpenter, 2003, reprinted with permission from Springer Nature).
**2.3 The relationship between Lightning channel electric field and current.**
The measurements of the differential resistance characteristics of a gas discharge gap on a
centimeter scale was conducted early by *King (1961)*. However, due to the effect of electrode
vaporization as pointed out by *Mazur & Ruhnke (2014),* King's results can only be applied to
currents less than 10A with short gaps. In larger-scale lightning channels, the current and electric
field are usually expressed in a power-law form. For instance, Bazelyan et al. (2008) assumed
that the leader channel current is inversely proportional to the electric field $E = 3400I^{-1}$, while



Larsson et al. (2005) suggested that the relationship between channel current and electric field
varies within the range of $10^2$-$10^4$A
$$E = 1600 I^{-0.18} \qquad (4)$$

This is consistent with the observations of Tanaka et al. (2003) and aligns with the

suggestions of da Silva et al. (2019) that the power law differs for each segment within the range
of $10^2$-$10^4$A. To better describe this relationship, we combined the data from King et al. (1961)
and Larsson et al. (2005).

For currents less than 10 A, we used the results of King et al. (1961). For currents greater

than 10 A, we applied the formula provided by Larsson et al. (Eq. 4).. Both sets of data were
fitted with a formula.
$$E = aI^b + cI^d \qquad (5)$$

Where $a = 4278, b = -0.9788, c = 1799, d = -0.2006,$ the minimum current for

fitting is taken to be approximately 0.1A.  Figure 3 shows the relationship between the electric
field and current, where the squares represent *King's* observations, the circles represent *Tanaka's*
*(2003)* experiments, and the solid green line represents the fit.

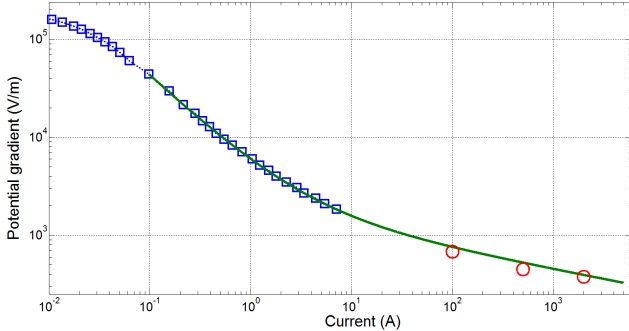


Fig3. electric field versus current in arc channel

**2.4 Differential resistance of the leader-streamer channel**

A streamer channel's resistance is determined by the potential difference $\Delta U_T$ of the

streamer zone of the leader head and the channel current $I$ , which can be expressed as (*Bazelyan*
*& Raizer, 2000)*
$$I = q_c V_L = 2\pi\varepsilon_0 V_L \Delta U_T \qquad (6)$$





where $q_c$ denotes the channel charge line density and $V_L$ denotes channel development speed, $V_L$
and $I$ follow a power-law relationship (Bazelyan & Raizer, 2000; Popov, 2009)
$$V_L = kI^{\alpha} \qquad (7)$$

As power exponents vary substantially among studies, for example $\alpha \approx 1/3$ (*Hutzler &*
*Hutzler, 1982, Bazelyan et al., 2007*), and $\alpha \approx 0.66$ (*Kekez & Savic, 1983*), in this paper, we
adopt $k = 1.88 \times 10^4$, $\alpha = 0.67$ based on more recent studies (*Andreev et al., 2008, Popov,*
*2009, Bazelyan et al., 2009*).
From Eqs. (6) and (7), we obtain the voltage drop in the streamer zone at the leader head
$$\Delta U_T = \frac{I}{2\pi\varepsilon_0 kI^{\alpha}} = \frac{I^{1-\alpha}}{2\pi\varepsilon_0 k} \qquad (8)$$

Considering the leader channel potential drop $U_C = LE$, where $L$ is the leader channel
length and $E$ is the electric field of the channel as shown in Eq. (4), and the streamer channel
potential drop $\Delta U_T$ as shown in Eq. (8), the total potential drop $U$ of the leader-streamer system
is as follows:

$$U = L(\mathrm{a}I^b + cI^d) + \frac{I^{1-\alpha}}{2\pi\varepsilon_0 k} \qquad (9)$$

Derive both sides with respect to $I$ gives the total differential resistance
$$\frac{dU}{dI} = L(\mathrm{ab}I^{b-1} + cdI^{d-1}) + (1-\alpha)\frac{I^{-\alpha}}{2\pi\varepsilon_0 k} \qquad (10)$$

**3 Analysis results**
Figure 4 shows how the differential resistance changes as the channel current increases for
different lengths of the leader channel, where the horizontal line represents zero differential
resistance. When the curve intersects the horizontal line, the differential resistance changes its
sign and the horizontal coordinate of the intersection indicates the critical current


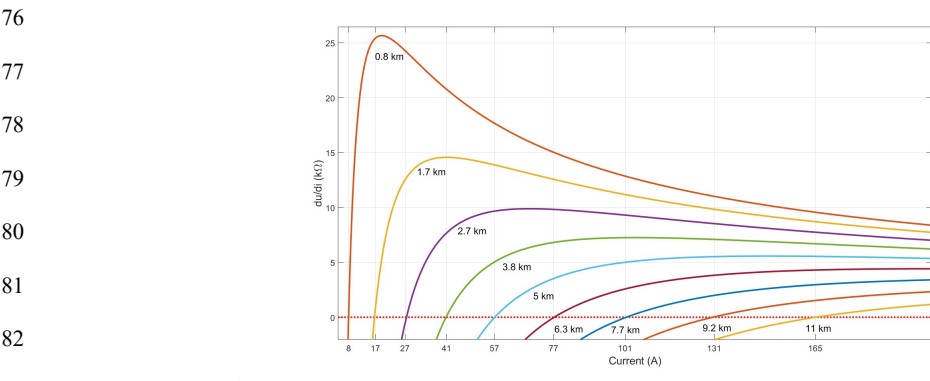

Fig4. Dependence of total differential resistance of channel on current with varying channel lengths. The
horizontal line represents zero resistance.

Figure 5 shows that critical currents increase with channel length, which is consistent with
*Heckman's 1992* study. It is also shown that the critical potential difference for the streamer zone
of the leader's head also increase with leader length, which aligns with the threshold condition
for the critical potential difference of the leader's development proposed by *Bazelyan & Raizer*
*(2000)* and *Mazur (2016a)*.

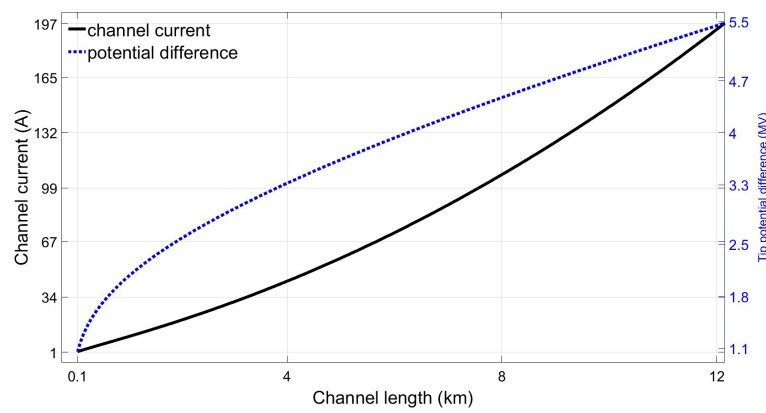

Fig5. Critical channel current and potential difference of the streamer channel at the leader tip vary with
channel length.

It can be observed that as the leader channel length increases, leader channel's ambient
(stabilized) electric field decreases, between 0.1 km and 12 km, the ambient electrical field of the
stabilized leader-streamer drops from 15.5 kV/m to 1.1 kV/m,  this is consistent with  *Lalande et*



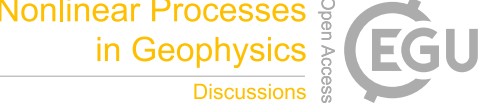

*al. (2002)* and *Becerra & Vernon (2006)* that the leader channel's ambient (stabilized) electric
field decreases with the channel's height. Similarly, the internal electric field of the leader
channel decreases(Figure 6). At a length of 0.1km, the electric field is about4.9 kV/m, while at
12km, it drops to 0.65kV/m, *Syssoev & Shcherbakov (2001)* determined that stable thermal
leader channels with long electric fields (30﹣50 m) were about 3﹣10 kV/m from laboratory
discharges, which are also similar to our results.

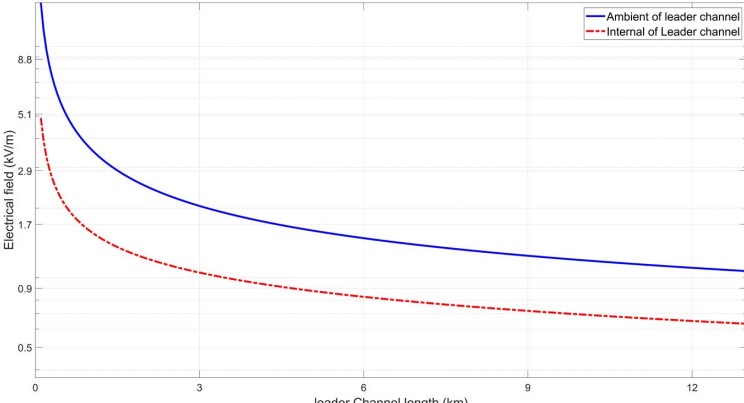


Fig 6. variations of ambient electric field of the leader-streamer system and electric field of the leader
channels with length
**4 Discussion and conclusion**

This paper extends the discussion of lightning discharge channel stability and channel

differential resistance from the leader channel to the leader-streamer system. Based on the
bifurcation theory and critical transition theory of nonlinear dynamics, the extinction, re-
excitation, and critical transition of intermittent events (such as recoil leaders) in the lightning
process were studied. By analyzing the sign changes in the differential resistance of the leader-
streamer system, the critical current and the critical potential difference in the streamer zone at
the channel end were obtained. The results show that as the channel length increases, the critical
current of the leader channel and the critical potential difference at the channel end also increase.
Meanwhile, the average ambient electric field and the channel electric field required for stable
transmission gradually decrease after an initial sharp drop. These findings are qualitatively
consistent with existing research results.





The specific mechanism behind the sudden change in channel conductivity remains
unclear but is undoubtedly related to the instability caused by positive feedback in the channel.
The re-excitation of a decayed leader channel is usually due to uneven distribution of current and
electric field. The development of a longer channel may exacerbate this inhomogeneity.
Typically, the leader head has a higher charge concentration and conductivity, making it more
active and persistent, often merging with adjacent channels. In contrast, the rear part of the
channel has relatively weaker conductivity and is more prone to disconnection and splitting. This
interaction of strength and weakness, merging, and splitting leads to the re-excitation of recoil
leaders.
In the case of negative ground flashes, the electric field in the upper channel becomes
non-uniform due to the low current in the positive leader section, which is insufficient to
maintain the conductivity of the lower channel. Recent observations (Williams & Montanya,
2019; Hare et al., 2019; Pu & Cummer, 2019; Hare et al., 2021) have found that the low current
in the positive leader and the poor conductivity of its corresponding rear leader result in negative
charge deposition in the center of the rear channel. This creates a series of outwardly directed
negatively polarized needle structures, triggering nonlinear instability. Consequently, the current
in the rear part of the positive leader decreases, causing it to disconnect from the negative leader.
The increased potential difference at the end of the paused negative leader results in its re-
breakdown and reconnection, forming multiple strokes in the negative ground flash process. In
the case of positive ground flashes, the upper part of the channel is a negative leader. The
stronger current at the head of the negative leader makes the channel less prone to splitting,
resulting in a single stroke.
The transition from a semiconductor to a conductor state in the leader channel may be due
to positive feedback caused by ionization-thermal instability in the channel. As shown in studies
by Bazelyan & Raizer (2000), Popov (2009), and da Silva et al. (2019), the pulsed mechanism of
the stepped leader is often related to the electric field inhomogeneity among the numerous
streamers at the head of the negative leader (Syssoev & Iudin, 2023). The triggering mechanism
may be attachment instability (Douglas-Hamilton & Mani, 1974), which exacerbates the
inhomogeneity of the electrical properties in the streamer zone (Sigmond, 1984; Luque et al.,
2016; Malagón‑Romero & Luque, 2019; Malagón Romero, 2021). If the mechanism of the
positive stepped leader is similar to that of the negative stepped leader, the stepped excitation



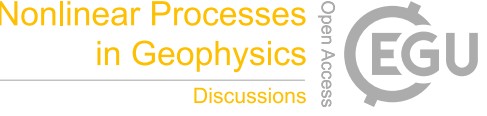

should occur in the streamer zone at the leader head (Tran & Rakov, 2016; Kostinskiy et al.,
2018; Huang et al., 2020; Wang et al., 2020).
Furthermore, whether in the initiation or transmission process, the various
inhomogeneities, instabilities, and critical transitions in the leader channel and streamer zone, as
well as the emergence of pulse events of different scales and interactions between leader
channels, streamers, and various streamers, all exhibit collective, fractal, and critical properties.
This may require more unified explanations based on fractal analysis and critical dynamics.
**Acknowledgments**
The work leading to this paper was supported in part by the National Natural Science
Foundation of China (Grant No. 42065005,42175100 and 42175090) and in part by The Hong
Kong Polytechnic University through the Research Grants Council of Hong Kong under Grant
GRF15215120
`Copyright and Permissions:`We affirm that all third-party copyrighted
material used in this manuscript has been obtained with proper permissions.
All such material is appropriately cited in the text and captions. Any
distribution licenses different from the standard CC BY (Creative Commons
Attribution) license are clearly stated below.
Figure 1: Adapted from "Experimental and Theoretical Investigations of the
Negative Differential Resistance in a Discharge Plasma," by Maricel Agop et
al., published in the Journal of the Physical Society of Japan. Used with
permission from the Physical Society of Japan. License: Non-commercial use
only.
Figure 2: Adapted from "Catastrophic regime shifts in ecosystems: linking
theory to observation," by Marten Scheffer and Stephen R. Carpenter, published
in Trends in Ecology & Evolution. Used with permission from Elsevier. License:
CC BY-NC.

## Code/Data Availability

The data and code used in this study are available from the corresponding
author on reasonable request.

## Author Contribution




- **Xueqiang Gou**: Conceptualization, Methodology, Software, Validation, Formal analysis, Investigation, Writing – Original Draft, Review & Editing, Visualization, Supervision, Project administration.
- **Chao Xin**: Methodology, Software, Formal analysis,
- **Liwen Xu**: Visualization.
- **Ping Yuan, Yijun Zhang, Mingli Chen:** Conceptualization, Review & Editing Supervision

## Competing Interests

The authors declare that they have no known competing financial interests or personal relationships that could have appeared to influence the work reported in this paper.

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
