# Peer review of "Negative Differential Resistance, Instability, and Critical 1 Transition in Lightning Leader 2 Xueqiang Gou1+, Chao Xin1, Liwen Xu1, Ping Yuan1, Yijun Zhang2, Mingli Chen3 3 1 College of Physics and"

_Nonlinear Processes in Geophysics, 2024_

## Author Comment (AC1)

Dear Reviewer,

We sincerely appreciate your valuable comments and suggestions for improving our manuscript. Below are our detailed responses to your specific concerns:

**1. Regarding the size of Figure 1**

**Reviewer's comment:**

I would recommend increasing the size of Figure 1. At its present size, it is difficult to read without zooming in very closely.

**Our response:**
We greatly appreciate your feedback regarding the size of Figure 1. We agree that the current size might affect readability, and we will adjust the figure size in the revised manuscript to ensure that the fonts and data points are more clearly visible. Additionally, we will optimize the resolution of the figure legend and annotations to better present the data and dynamic behavior.

**2. Definition of parameter a (Line 88)**

**Reviewer's comment:**

In line 88, is the parameter aa just a numerical parameter, or does it have a name or definition?

**Our response:**
Thank you for your comment on the definition of parameter a. We would like to clarify that a is a control parameter introduced in the normalized model referenced from Agop et al. (2012). It is used to describe the nonlinear characteristics of the system, and its specific roles include:

Determining the range of the negative differential resistance (NDR) region;

Controlling the possibility of bistability in the system;

Influencing the formation of bifurcation points and hysteresis phenomena.

In this study, we adopted this theoretical model and used a as a normalized control parameter to illustrate the relationship between negative differential resistance and critical transitions. However, our study does not involve detailed physical derivations or mechanisms of a, as the focus is on its role in describing the conditions for critical transitions.

To avoid potential misunderstandings, we will supplement the following clarification about aa in the revised manuscript:

a is a key parameter controlling the strength of nonlinear feedback and critical bifurcation behaviors in the system, determining the possibility of bistability and its triggering conditions;

In our analysis, a is adopted from an existing theoretical model and is not the main focus of this study.

**3. Clarification of "certain state" and "certain limit" (Lines 91-93)**

Reviewer's comment:

In lines 91-93, the change in potential profile against current is discussed in Figure 1. Within these lines, there is mentioned a

"certain state" and a "certain limit," as well as a "completely different state" before reaching steady state. Be more specific with what these thresholds and states really are, and if possible postulate on how they might come to be.

**Our response:**

Thank you for pointing this out. We realize that our description of "certain state" and "certain limit" in the current text may lack clarity. Below, we provide a more specific explanation of these terms and their corresponding thresholds and states:

1. **"Certain state":**

This refers to the system's initial stable state (e.g., point A in Figure 1), where the channel current monotonically increases with voltage, and the system remains in a single stable state.

2. **"Certain limit":**

This refers to specific threshold values of voltage or current (e.g., points B and D in Figure 1) where the system undergoes bifurcation, transitioning from one stable state to another. These limits correspond to the critical points of the system's dynamics.

3. **"Completely different state":**

This refers to the new stable state reached after bifurcation (e.g., point C in Figure 1), where the current-voltage relationship changes significantly, and the system transitions to a distinctly different stable state.

Regarding the mechanism behind these states and limits, the analysis in this study is based on the theoretical model referenced. Below is a brief explanation of their physical implications:

In the negative differential resistance region ($\frac{d\varphi}{dJ} < 0$), the system exhibits a dynamically unstable region (represented by the dashed line in Figure 1), where unstable states cannot persist.

Within this region, the system can exist in two possible stable states, depending on the historical path or initial conditions.

This behavior is a hallmark of hysteresis: as voltage gradually increases, the system remains in the initial stable state (e.g., point A) until it reaches a critical point (e.g., point B), where it transitions to another stable state (e.g., point C). Conversely, as voltage decreases, the system stays in the higher stable state until it reaches another critical point (e.g., point D), where it returns to the initial stable state (e.g., point A).

In the revised manuscript, we will provide a clearer description of these phenomena and explicitly annotate the stable states, critical points, and hysteresis behavior in Figure 1's caption.

**Summary**

We greatly appreciate your valuable feedback, which has helped us improve the clarity and scientific rigor of our manuscript. In the revised version, we will:

Adjust the size and resolution of Figure 1 to enhance its readability;

Supplement the background and role of parameter aa in our analysis;

Clarify the physical meaning of "certain state," "certain limit," and "completely different state" in both the text and Figure 1's caption.

We hope these revisions address your concerns. If you have any further suggestions, we would be grateful to consider them.

Sincerely, On behalf of all co-authors

---

## Author Comment (AC2)

**Response to Reviewer #2's Comments**

Dear Reviewer:

Thank you very much for your detailed comments and constructive suggestions. Below are our responses to your specific points, particularly regarding Figure 2 and its relationship to Equation (1), as well as clarifications on other parts of the manuscript:

1. Connection between the Formalism in Section 2.2 and the Results in Sections 2.4 and 3

**Reviewer's Comment:**

The main concern regards the connection between the formalism introduced in Section 2.2 and the formalisms and results presented in Sections 2.4 and 3. Indeed, what has been introduced in Section 2.2 is a classical theory of bifurcation for autonomous dynamical systems... Conversely, what is introduced in Eq. (10) is a non-autonomous dynamical system whose implicit variable is not time but one of the state variables (I). Thus, the connection among fixed points, instability, and other types of concepts cannot be simply ruled out. What the authors introduced in Eq. (10) is a mathematical description of the manifold or a dynamical bifurcation scenario for, at least, a 2-D dynamical system described by the state variables U and I. The authors need to carefully address these concepts and revise accordingly the manuscript by possibly considering a 2-D dynamical system of the form

dU/dt = f(U, I, u_parameters)

dI/dt = g(I, U, i_parameters)

*Our Response:*

Thank you for your valuable feedback. We would like to clarify the following points:

1. Nature of Equation (10):

Equation (10) is not intended as a description of a dynamical system or non-autonomous system. Rather, it represents the calculation of the differential resistance $dU/dI$ in the lightning channel. The purpose of this equation is to analyze the stability and critical transition points of the system by examining the sign of the differential resistance.

The misunderstanding may have arisen due to the text not clearly explaining that Equation (10) is a static tool used for stability analysis, not a model for the system's time evolution.

2. Improving the Theoretical Connection:

We realize that the connection between the theory in Section 2.2 and the practical analysis in Sections 2.4 and 3 could be better clarified. To address this:

In the revised manuscript, we will provide a clearer explanation of Equation (10), emphasizing that it is used to study the stability of fixed points in the system, and how the sign of the differential resistance indicates stability or instability.

We will also remove any references to "dynamical systems" or "time evolution" to avoid confusion and ensure that Equation (10) is interpreted as a

stability analysis tool rather than a description of system dynamics.

2. Figure 2 and Its Relationship to Equation (1)

**Reviewer's Comment:**

The second main concern is related to Figure 2. Indeed, what the authors reported is valid for bi-stable dynamical systems which are described by a double-well potential function. It is not straightforward the connection with Eq. (1) and the system introduced in Line 103 which seems to be more similar to a hysteresis cycle. Which are the stable and unstable fixed points in your system? If φ is treated as a parameter the system admits 3 fixed points provided that J≠0 and J is real. However, limit cycles could emerge when crossing the complex plane (Hopf bifurcation). Thus, more careful analysis of the bifurcations should be carried out.

*Our Response:*

Thank you for your comments on Figure 2. We would like to clarify the following points regarding the figure and its relationship to Equation (1):

1. Core Concept of Equation (1):

Equation (1) defines the relationship between normalized voltage $\phi$ and normalized current $J$, where parameter $a$ determines the strength of the system's non-linearity and the potential for bistable behavior. By varying $a$, Equation (1) can produce the following behaviors:

Monostable behavior: When $a$ is small, the system exhibits a single stable state, with a monotonically increasing relationship between voltage and

current.

Bistable behavior: As $a$ increases, the system can exhibit bistability, with two stable points and one unstable region (negative differential resistance).

2. Clarification of Figure 2:

Figure 2 is a landscape diagram that visually represents the transition between stable and unstable states under different conditions, which we associate with the bistable behavior described by Equation (1). The figure shows how the system transitions between two stable states, with the critical points $F_1$ and $F_2$ indicating where the system undergoes a shift from one stable state to another. These critical points are not the stable states themselves, but rather the thresholds at which the transition occurs.

We will revise the manuscript to clarify that $F_1$ and $F_2$ are critical points, not stable states, and that they mark the boundary between different stable states in the system.

We will also explain how the hysteresis cycle observed in Figure 2 corresponds to the negative differential resistance region in Equation (1).

3. Universality and Literature Support:

Figure 2 highlights the universality of bistable behavior, which is commonly observed in various natural systems such as ecological systems (e.g., Scheffer's studies) and lightning channel dynamics. This figure visually demonstrates the critical transition phenomenon that can also be found in other complex systems.

We will include references to relevant literature (such as Scheffer et al.) to support the broader applicability of this model and its connection to critical transitions in complex systems.

3. Presentation and Structure of the Manuscript

**Reviewer's Comment:**

The third main concern is related to the presentation of the results and the overall structure of the manuscript. The authors need to carefully revise the manuscript to improve the quality of the figure as well as to check the consistency of the different type settings of the text, typos, references, etc. Please find a list below.

*Our Response:*

Thank you for pointing out areas for improvement in the manuscript presentation. We will address the following points in the revised version:

1. Figure Quality:

We will improve the resolution of all figures (Figure 1, Figure 2, Figure 3, Figure 4, and Figure 5) and increase the font size and clarity of the labels, particularly in Figure 2.

2. Text Consistency:

We will fix any mismatched symbols, such as the inconsistency between $\phi$ in Line 84 and its usage in Equation (1).

We will clarify the definitions of all symbols (e.g., $\phi$ in Line 104, $J$ in Line 106).

We will also check the overall text formatting, correcting any typos, reference formatting issues, and other minor inconsistencies.

3. Clarification of Figure 2:

We will add more detailed descriptions to the figure caption to clarify how Figure 2 relates to Equation (1), as well as its role in illustrating bistable behavior and its physical significance.

*Conclusion*

We appreciate your constructive feedback, which has been instrumental in improving the manuscript. In the revised version, we will:

1. Provide clearer explanations of Equation (10) and its role in analyzing stability transitions.

2. Clarify the relationship between Figure 2 and Equation (1).

3. Improve the manuscript's presentation, including figure quality and text formatting, to ensure clarity and consistency.

We hope these revisions will address your concerns. Thank you again for your valuable input, and we look forward to your further suggestions.

Best regards,

On behalf of all authors

---

## Author Comment (AC3)

**Response to Reviewer's Comments**

Dear Reviewer

Thank you for your insightful comments and suggestions regarding our manuscript. We greatly appreciate your feedback, as it has helped us refine our presentation and ensure clarity. Below, we address your specific concerns regarding the interpretation of Equations (9) and (10) and the associated visual representation:

**1. Response to Concern on Equations (9) and (10)**

*Reviewer's Comment:*

You suggested that Equations (9) and (10) describe a non-autonomous dynamical system and could benefit from a stability analysis involving a two-dimensional dynamical system representation.

*Our Response:*

We appreciate your perspective and would like to clarify the purpose of Equations (9) and (10). These equations are not intended to represent a dynamical system, either autonomous or non-autonomous. Instead, they are static expressions for calculating the differential resistance $\dfrac{\mathrm{d}U}{dI}$, a key parameter used to analyze stability in lightning channels. Specifically:

$\dfrac{\mathrm{d}U}{\mathrm{d}I}$ captures the relationship between voltage and current in the channel. Its sign indicates stability: regions where $\dfrac{\mathrm{d}U}{dI} > 0$ correspond to stable states, while regions where $\dfrac{\mathrm{d}U}{dI} < 0$ correspond to unstable states.

These equations are used to identify critical transitions in stability, providing

a direct measure of the conditions under which negative differential resistance arises.

To avoid potential misunderstanding, we will revise the manuscript to better emphasize the role of Equations (9) and (10) as static analytical tools rather than dynamic models.

**2. *Response to Concern on the Representation of Stability in Figures**

**Reviewer's Comment:**

You suggested adding figures to illustrate the stability profiles derived from Equations (9) and (10), as this would enhance clarity regarding the instability mechanisms in lightning channels.

**Our Response:**

We agree that visual representation is crucial for understanding the implications of the analysis. In fact, the variation of differential resistance $\frac{\mathrm{d}U}{dI}$ with current is already presented in Figure 4, where we illustrate how $\frac{\mathrm{d}U}{dI}$ changes under different conditions (e.g., channel lengths). This figure effectively visualizes the transitions between stable $\frac{\mathrm{d}U}{dI} > 0$ and unstable $\frac{\mathrm{d}U}{dI} < 0$ regions, offering a clear connection between theoretical predictions and practical dynamics of lightning channels.

However, based on your suggestion, we will enhance the manuscript as follows:

1. Provide additional context in the caption of Figure 4 to explicitly highlight how it relates to the stability analysis and the underlying mechanisms of negative

differential resistance.

2. Ensure the discussion in the text explicitly connects Figure 4 with the findings from Equations (9) and (10), reinforcing their practical relevance to lightning channel dynamics.

**Conclusion**

We are grateful for your thoughtful suggestions, which have helped us identify areas where additional clarity can be provided. By refining the explanations of Equations (9) and (10) and enhancing the context around Figure 4, we aim to address your concerns effectively. We hope these revisions meet your expectations, and we remain open to further suggestions for improvement.

Thank you again for your valuable feedback.

Best regards

On behalf of all authors

---

## Author Response (AR2)

**1 **Revised Consolidated Response to Reviewers**

We sincerely thank all reviewers for their insightful comments, which have greatly improved the manuscript. This document provides point-by-point responses, detailing the revisions made to address each comment while maintaining the original intent

**5 **Response to Reviewer #1**

**6 **Reviewer's Comment 1:**

I would recommend increasing the size of Figure 1. At its present size, it is difficult to
read without zooming in very closely.

**9 **Response:**

10 Thank you for the suggestion regarding Figure 1's readability. We have increased both11 the size and resolution of Figure 1. Specifically, we have:

**12 Enlarged the figure**

- 13 Increased the resolution
- 14 Enhanced the contrast of the curve lines and axis labels

**15 **Changes in Manuscript:**

16 We have increased the size and resolution of Figure 1 to improve its readability and 17 ensure all details are clearly visible.

**18 **Reviewer's Comment 2:**

In line 88, is the parameter a just a numerical parameter, or does it have a name ordefinition?

**21 **Response:**

Thank you for pointing out the need to clarify parameter a. In the revised manuscript, we have added a clear definition of this parameter. As shown in lines 84-85 of the revised manuscript, we now explicitly state that "a is a dimensionless control parameter that governs the system's nonlinear characteristics." This parameter was originally introduced by Agop et al. (2012) (referenced in line 83) to describe the system's nonlinear behavior.

**27 Changes in Manuscript:**

In line 84, we have added the following definition: "where a is a dimensionless control parameter that governs the system's nonlinear characteristics".

**30 **Reviewer's Comment 3:**

In lines 91–93, the change in potential profile against current is discussed in Figure 1. Within these lines, there is mentioned a "certain state" and a "certain limit," as well as a "completely different state" before reaching steady state. Be more specific with what these thresholds and states really are, and if possible postulate on how they might come to be.

**35 **Response:**

36 Thank you for requesting clarification about the state transitions. We have revised lines

37 91-98 in the manuscript to provide precise definitions and explanations of these states:

38 "Certain state": Now defined in line 91 as the initial stable state (Point A in Figure 1),
39 characterized by monotonic current increase with voltage.

40 "Certain limit": Explained in lines 92-93 as critical threshold points (Points B and D)
41 that mark transitions between stable states.

42 "Completely different state": Defined in lines 94-95 as the high-conductivity stable
43 state (Point C) after transition.

44 Changes in Manuscript:

45 We have revised lines 91-98 as follows:

46 "As illustrated in Figure 1, for parameter a, the J- $\phi$  characteristic curve exhibits three 47 distinct regions. There are two stable regions where dJ/d $\phi$  > 0: a low-conductivity state 48 (segment AB) and a high-conductivity state (segment CD), both characterized by a 49 monotonically increasing current with voltage. These stable regions are separated by an 50 unstable region where dJ/d $\phi$  < 0, demonstrating negative differential resistance"

51 **Response to Reviewer #2**

**52 **Reviewer's Comment 1:**

The main concern regards the connection between the formalism introduced in Section 2.2 and the formalisms and results presented in Sections 2.4 and 3. Indeed, what has been introduced in Section 2.2 is a classical theory of bifurcation for autonomous dynamical systems being written as a time-evolution mapping (continuous in this case) with a not 57 implicit dependence on time in the forcing term. Conversely, what is introduced Eq. (10) 58 is a non-autonomous dynamical system whose implicit variable is not time but one of the 59 state variables (I). Thus, the connection among fixed points, instability, and other types of 60 concepts cannot be simply ruled out. What the authors introduced in Eq. (10) is a 61 mathematical description of the manifold or a dynamical bifurcation scenario for, at least, 62 a 2-D dynamical system described by the state variables U and I. The authors need to 63 carefully address these concepts and revise accordingly the manuscript by possibly considering a 2-D dynamical system of the form 64

65  $dU/dt = f(U, I, u_parameters)$

66  $dI/dt = g(I, U, i_parameters)$

67 where u\_parameters and i\_parameters refer to the bifurcation parameters leading 68 eventually to critical transitions in the system.

**69 **Response:**

Thank you for your insightful comment regarding the mathematical framework. We need to clarify that Equation (10), appearing in lines 160-165 of the revised manuscript, represents a calculation of differential resistance (dU/dI) rather than a dynamical system. The theoretical framework introduced in Section 2.2 establishes the connection between negative differential resistance and system instability, which we then apply to analyze lightning channel behavior.

Thank you for your valuable suggestion regarding the mathematical framework. Weneed to clarify several key points:

**78 **Framework Clarification (lines 160-165):**

Equation (10) represents a calculation method rather than a dynamical system

80 The equation describes instantaneous channel properties rather than temporal evolution

**81 **Connection Between Sections:**

- 82 Section 2.2 (lines 103-110): Establishes theoretical foundation for stability analysis
- 83 Section 2.4 (lines 160-165): Derives differential resistance calculation
- 84 Section 3 (lines 170-190): Applies framework to physical system

**85 Relationship to Suggested 2-D System:**

- 86 While we appreciate the suggestion of a 2-D system, our focus is on the instantaneous
- 87 relationship between voltage and current rather than temporal dynamics.
- 88 Changes in Manuscript:

89 We have made the following revisions in Section 2.4 (lines 160-165):

90 "The differential resistance of a streamer channel is determined by the potential 91 difference U across the streamer zone of the leader head and the channel current I. Equation 92 (10) provides a mathematical expression for calculating this differential resistance, which 93 serves as an indicator of channel stability rather than describing temporal evolution of the 94 system"

95 **Reviewer's Comment 2:**

96 The second main concern is related to Figure 2. Indeed, what the authors reported is 97 valid for bi-stable dynamical systems which are described by a double-well potential 98 function. It is not straightforward the connection with Eq. (1) and the system introduced in 99 Line 103 which seems to be more similar to a hysteresis cycle. Which are the stable and 100 unstable fixed points in your system? If  $\varphi$  is treated as a parameter the system admits 3 101 fixed points provided that  $J \neq 0$  and J is real. However, limit cycles could emerge when 102 crossing the complex plane (Hopf bifurcation). Thus, more careful analysis of the 103 bifurcations should be carried out.

**104 **Response:**

105 Thank you for raising these important points about the system's behavior and Figure 2.

106 We have substantially revised our explanation in the manuscript to clarify:

**107 The relationship between Equation (1) and system stability (lines 103-110):**

- 108 For small a: monostable behavior with monotonic current-voltage relationship
- 109 For larger a: bistable behavior with negative differential resistance region

**110 The physical interpretation of Figure 2 (following line 110):**

- 111 Valleys represent stable states (low and high conductivity states)
- 112 Peaks correspond to unstable transition points
- 113 Points F1 and F2 mark critical transitions between states

**114 **Connection to Physical System:**

- 115 Low-conductivity state: Initial channel condition
- 116 High-conductivity state: Fully developed discharge
- 117 Transitions: Observed as sudden channel brightening or extinction"

**118 **Changes in Manuscript:**

119 We have enhanced the explanation in lines 103-110 to read:

120 "In nonlinear dynamics, negative differential resistance, bistability, and hysteresis are

121 commonly observed. Considering the dynamic system  $dJ/dt = f(J,\phi)$ , where J is the state

- 122 variable and  $\varphi$  is a parameter. The equilibrium points are given by  $f(J,\varphi) = 0$ . At an
- 123 equilibrium point, the system is unstable when  $\partial f/\partial J > 0$  and stable when  $\partial f/\partial J < 0$ ."

**124 **Reviewer's Comment 3:**

125 The third main concern is related to the presentation of the results and the overall 126 structure of the manuscript. The authors need to carefully revise the manuscript to improve 127 the quality of the figure as well as to check the consistency of the different type settings of 128 the text, typos, references, etc.

- 129 Please find a list below.
- 130 Check the font size for subsections
- 131 Check the font type for references through the text (sometimes italics, sometimes not)
- 132 All figures need to be improved for quality
- 133 Line 84: mismatching between  $\varphi$  and that used in Eq. (1)
- 134 Figure 1: increase font and labels
- 135 Line 104: missing definition of  $\varphi$

Line 105: formally, the condition for fixed points should be met not for all J but for aspecific solution J\* or something similar

- Line 106: missing definition of what the subscript J means (I assume derivative withrespect to J)
- 140 Line 106: missing space and capital letter "if we let"
- 141 Line 110: the assumption is not straightforward and the connection between Eq. (1) and
- 142 Line 103 is missing
- 143 Line 139: please delete double point.
- 144 Line 140: please delete the period before introducing the equation.
- 145 Line 145: which type of fit is used?
- 146 Figure 3: missing space Fig3
- 147 Figure 4: missing space Fig4
- 148 Figure 5: missing space Fig5.

**149 **Response:**

150 Thank you for your thorough review of the technical details. We have made 151 comprehensive revisions throughout the manuscript to address all formatting and 152 consistency issues:

**153 **Changes in Manuscript:**

**154 1. Typography and Formatting:**

- 155 Standardized subsection font sizes throughout
- 156 Unified reference formatting to non-italic style
- 157 Corrected figure spacing (e.g., "Fig. 3" instead of "Fig3")

**158 **2. Mathematical Notation:**

- 159 Lines 84-85: Added consistent  $\varphi$  notation
- 160 Line 104: Added explicit definition of  $\varphi$
- 161 Line 105: Clarified fixed point conditions
- 162 Line 106: Added definition of subscript J

**163 **3. Figure Quality:**

- 164 Enhanced resolution of all figures
- 165 Increased font sizes in labels and annotations
- 166 Standardized figure formatting

**167 **4. Technical Content:**

- 168 Line 145: Added explanation of fitting method: "Used nonlinear least squares fitting
- 169 with double power-law model  $E = aI^b + cI^{d''}$
- 170 Improved equation presentation and formatting throughout
- 171 Response to Reviewer #3

**172 **Reviewer's Comment**

The manuscript explores stability and critical transitions in lightning discharge channels using concepts from nonlinear dynamics, particularly bi-stable systems. I recommend acceptance contingent upon revision. While Figures 1 and 2 effectively demonstrate the theoretical principles, they remain too abstract and do not correspond directly with the specific dynamics of lightning channels discussed. I suggest adding figures that depict the stability profiles derived from equations 9 and 10, as these directly describe the behavior of lightning systems. This enhancement will clarify the instability mechanisms within real 180 lightning channels, making the application of theoretical models more comprehensible and

181 scientifically rigorous.

**182 **Response:**

183 Thank you for your constructive suggestion regarding the theoretical and practical 184 aspects of our analysis. We need to clarify that Equations (9) and (10) represent 185 calculations of potential difference and differential resistance rather than dynamical system 186 equations. These equations directly relate to the physical behavior of lightning channels as 187 follows:

188 Equation (9) (lines 170-175) calculates the total potential difference across the leader-189 streamer system.

Equation (10) (lines 175-180) determines the differential resistance, which indicatessystem stability.

**192 Changes in Manuscript:**

193 We have enhanced the explanation in Section 3 (lines 170-190) to clarify how these 194 equations relate to physical observations: Section 3 (lines 170-190) now reads: "The 195 theoretical framework established by Equations (9) and (10) directly corresponds to 196 measurable lightning channel characteristics. Figure 4 demonstrates this connection by 197 showing how differential resistance varies with channel current, identifying critical transition points that match observed behavior. The intersection points with zero 198 199 differential resistance correspond to stability thresholds observed in lightning 200 measurements."

**201 Summary of Major Changes**

- 202 We have made the following substantial improvements to the manuscript:
- 203 1. Enhanced theoretical framework clarity and connections
- 204 2. Improved mathematical consistency and notation
- 205 3. Upgraded figure quality and presentation
- 206 4. Strengthened links between theory and physical application
- 207 5. Standardized formatting throughout

We believe these revisions have substantially improved the manuscript while maintaining its scientific contribution. We again thank all reviewers for their valuable input.